# Swine Breeding in the Villages of Vâlcea County, Oltenia (Romania)—Tradition or Necessity?

Camelia Teodorescu [1], Marin Burcea [2], Ana-Irina Lequeux-Dincă [1,*], Florentina-Cristina Merciu [1], Adrian-Nicolae Jipa [1] and Laurenţiu-Ştefan Szemkovics [3]

[1] Faculty of Geography, University of Bucharest, 1. Blv. Nicolae Bălcescu, 010041 Bucharest, Romania
[2] Faculty of Administration and Business, University of Bucharest, 4-12 Regina Elisabeta Av., 030018 Bucharest, Romania
[3] National Central Historical Archives, 49 Regina Elisabeta Av., 050013 Bucharest, Romania
* Correspondence: ana.dinca@geo.unibuc.ro; Tel.: +40-726-691-727

**Abstract:** Food supply has been a constant source of concern for mankind. In the present context, with food security a priority of European and national policies, an analysis of pig farming in a representative NUTS2 administrative level of Romania that emphasizes the proportion of households raising at least one pig and the main factors influencing farmers to adopt or give up swine breeding could allow a much clearer understanding of this phenomenon that lies at the border between cultural tradition and socio-economic necessity. This study uses mixed methods that complement each another to help reveal this complex phenomenon in the analyzed territory. Cluster analysis shows the concentration of swine breeding and maps its spread in terms of both subsistence and larger farms, and qualitative interviews prove the motivation of farmers to continue in this occupation. As a primary result, the study visualizes the spatial distribution of pig farming in the rural environment of Vâlcea county, Romania, from a diachronic perspective in the post-communist period. It also reveals areas of differing concentrations of both very small-sized farms, which prioritize meeting their own food needs, and larger farms, which prioritize commercial production to supplement their revenue streams. Both categories, but particularly the latter, are of particular interest in a period in which the socio-economic environment after 1990—marked by economic restructuring, unemployment, population migration, the economic crisis of 2008–2010, the pandemic of 2020–2021, and the most recent energy crisis—periodically highlights the importance of rural areas in ensuring food security and sufficiency at both the local and regional levels.

**Keywords:** animal breeding; economic crisis; local culture; rural economy; cluster analysis

## 1. Introduction

Swine and animal breeding are aimed at meat consumption [1,2]. Meat consumption, including pork, has reached notably high values [3,4], with the trends reflecting the growing increase in meat consumption [4,5], as meat is an essential food that serves as an important source of protein and fat [2,6,7]. The OECD-FAO (2021) report [8] estimates an increase of 14% in the global consumption of meat proteins this decade compared to average consumption in 2018–2020 driven by urbanization and income and population growth [4]. Protein consumption from pork is projected to increase by 13.1%. This protein source is the third-most consumed after beef (estimated to grow by 17.8%) and lamb (estimated to grow by 15.7%) [8]. In the European Union, the average consumption of pork is 41 kg/person, which is 3.28 times higher than the average global consumption (12.5 kg/person) [9]. The EU is also the biggest producer of pork after China and the biggest exporter of pork and its associated products. Its exports were recently boosted by a decrease in production in Asia due to African swine flu, which led to a price peak for pork products in early 2020 [10]. According to the same European Parliament briefing report, pigs represent the largest

livestock category across the EU, even more than bovines, and account for nearly half of the total EU meat production.

Studies have confirmed that Eastern Europeans are traditional meat eaters and researchers have observed an increase in the production of pig meat [4,11] and poultry on the European meat market [11] compared to other consumer patterns that show preferences for beef or lamb.

In Romania, meat consumption has increased in recent years from 54.4 kg/person in 2013 to 76.7 kg/person in 2021. Pig meat is the most frequently consumed meat product, with about 50% of the total national meat consumption (38.3 kg/person), surpassing poultry (27 kg/person) [12,13].

When considering the importance of swine breeding [14], this activity, along with plant and cereal cultivation [15–17], can help to meet the food needs of a continuously growing world population and achieve food security [17–21].

Finding food sources has been of great concern to humans for a long time [22]. Domestic pigs spread throughout central and southern Europe around the Mediterranean Sea and along large European rivers, reaching territories to the north of the Danube River in 5500 BC [23]. According to different sources, swine breeding has existed within the Romanian territory since ancient times [24].

Romania has an interesting history of agricultural occupation and market development within the EU. It is one of the primary pork-consuming countries in the EU, with a strong tradition of homemade pork production and a wide array of pork-based dishes, although the per-capita consumption is slightly lower than the EU average (38.5 kg/person compared to 41 kg/person) [25].

Pig livestock numbers in Romania have decreased significantly in the last few decades, from about 12 million in 1990 to 4.79 million in 2000, decreasing to 3.92 million in 2018, and then to 3.54 million in 2021 according to the National Institute of Statistics press release no. 232 from 15 September 2022 [26]. This is in contrast with other countries in the EU, which have more intensive production (including Spain, Germany, France, Denmark, and the Netherlands). However, the EU is the main source of pork imports for Romania, which maintains high consumption and demand of these products on the local market, thus making them an important part of the EU market. According to studies of Eastern European meat consumer patterns, a significant percentage of the Romanian population consumes pork on a daily or weekly basis [11].

The cultural aspects should not be neglected, as raising household pigs and slaughtering them on Ignat Day (20 December) (around the winter solstice) has been a ritual in Romania since the pre-Christian era. Furthermore, this practice is associated with the end of the Advent season and the days leading up to Christmas, and over time, has given rise to a variety of complex traditions and laic rituals [27].

When considering the essential role played by agriculture in providing food security, the present research analyzes the topical issue of pig farming in Vâlcea County, Oltenia Province, Romania. In this region, pig farming is part of the local culture and has a long tradition. Moreover, there has been an increasing trend toward pig farming, especially in rural settlements. The main objective of this study is to thoroughly analyze the main motivations for swine breeding at the level of existing farms in each settlement located in Vâlcea County in relation to the main factors that influenced this activity between 1990 and 2022. The study applied both quantitative and qualitative research methods to achieve a complex, in-depth analysis of swine breeding in the selected territory. Ample face-to-face interviews were used to understand what influenced people's choices to breed swine within the communities in Vâlcea County. Statistical calculations based on data gathered through ample field surveys were used to establish the number of households involved in swine breeding out of the total number of households in the case study area.

This paper focuses on the following research questions:

1.  What is the proportion of households breeding at least one pig in each rural administrative unit (commune) in Vâlcea county in the four selected years of 1990, 2000, 2010 and 2022 based on the existing demographic and economic resources ?
2.  What are the main factors that influenced farmers to adopt or give up swine breeding in their household?

The present study could, therefore, serve as a valuable contribution to the existing literature, as it analyzes the factors influencing swine breeding in a region where agriculture is the dominant occupation. The relevance of this research is underscored by its focus on swine breeding as an efficient solution to food and economic crises and its emphasis on the role played by agriculture in providing food security. This paper may, consequently, support similar analyses in other rural areas.

## 2. Literature Review

When analyzing the scientific literature on the topic of swine breeding and its specific characteristics in Romania, several perspectives are evident. One is the examination of the importance of swine breeding as a source of food and nutrients [1,2] and the role of fresh pig meat in human nutrition [2,4,7,9]. Domestic pigs represent the second-most important global source of meat [28].

The growing world population needs adequate sources of protein to maintain food security [20], and undoubtedly, proteins of animal origin will remain an important part of the global food system [25,29]. Consequently, animal breeding is estimated to intensify in the future [25]. The demand for high-quality proteins has influenced the design of several innovative solutions in recent years (e.g., the use of algae for animal feed to increase the quality and nutritional value of pork and poultry) [30]. Some studies have highlighted recent efforts toward conventional rotational cropping [31], as well as improvements in cropping techniques and fertilization to increase both yields and crop health [16,32]. Other studies have proposed viable solutions such as reductions in supply chains [15], particularly in the current context in which the Russian invasion of Ukraine affects global food security, [33] as well as for certain African countries that are dependent on cereal imports [21,34]. To increase meat production, there has been a focus on the genetic improvement of pigs through selective breeding aimed at enhancing meat output and resistance to diseases, ultimately leading to greater efficiency in swine breeding [35–37].

Another research perspective focuses on environmental protection, including ecologically sustainable solutions for animal and swine breeding, and on diseases that can affect animals and result in critical economic, social, and ecological damage [1,14,28,38].

A third perspective focuses on the economic importance of swine meat and its consumption, as well as on the breeding and valuation of pig livestock [3–6,11]. Several studies have focused on an analysis of the demographic factors of swine breeding activities [11,39].

In Romania, swine farming has been influenced by a number of factors. Among them were the profound economic and political changes in 1990, which brought about significant transformations regarding land reform [22,40,41]. In addition, the "adaptation of the production structures to the new free economy requirements" should not be overlooked p. 293, Ref. [40].

The profound political and socio-economic changes also influenced the emergence of free trade and the access of the Romanian population to products other than local ones, especially products from European markets. These changes also led to periods of economic recession (e.g., 2008–2010), oscillating rates of inflation, population aging, and high rates of employment in agriculture, particularly in rural areas. All these factors influenced the trajectory of individual households and greatly impacted swine farming in rural environments in Romania [42].

The present context also induced other supplementary, important economic stress factors, such as refugee crises (faced by the EU and Romania, in particular, in 2016 but also at the present moment because of the conflict in the nearby region) [43], the SARS-CoV-2 pandemic, and the energy crisis, as well as repetitive health crises such as African swine

flu, the most recent outbreak of which was confirmed in October 2022 [44], which is of particular concern to pig farmers in Romania. The great economic losses due to swine flu influenced government measures that materialized through legislative acts such as Government Decision no. 830/2016 [45], which introduced supplementary measures to prevent the occurrence of African swine flu in Romania, and again emphasized the importance of swine breeding in the local landscape.

At the same time, the growing number of pigs in rural farms is influenced by economic necessity [46,47] and is permitted due to the privatization and liberalization of the agricultural land market in both rural regions and on the outskirts of urban areas in Romania [48] since the fall of the communist regime in 1990. The threshold between tradition and economic necessity for agricultural occupations is influenced by other factors, such as the development of the local economy, the employment rate, the income per household, and the average age of inhabitants in each settlement. Studies [39,49] show that between 2018 and 2019, the employment rate in agriculture was generally high, with spikes observed in the Southern and Eastern counties. Moreover, the excessive fragmentation and division of farms in Romania's agricultural holdings into multiple small plots have led to a high prevalence of subsistence and semi-subsistence farming, with small-scale family farms focusing on both social and economic benefits [50].

A challenge for small farms is to cover the target level for food self-sufficiency through their meat production. Another challenge for rural development in Romania is the significant aging of the rural population [51], which underscores the need to prioritize this domain in local development efforts by identifying viable paths for growth [52]. According to some studies, remote rural areas have a high percentage of elderly residents who are particularly vulnerable due to a lack of essential services that address their current needs [53].

After 1990, important socio-economic transformations in Romania influenced the fluctuating relationship between consumption, animal breeding (especially swine), and family savings. Studies show that in the post-socialist period, consumption became a priority and savings rates even became negative between 2000 and 2010 [52]. This phenomenon can be explained by both the decrease in the number of livestock and the influence of demographic factors (e.g., farmers' aging). After 2010, new modern, small-sized rural farms for swine breeding appeared. Private savings and bank loans were mainly used to fund very small farms (5–10 pigs), whereas bank loans and sometimes European structural funds were used to fund small farms (more than 11 pigs). Both categories, especially the latter, are expected to contribute to the development of livestock production in local agriculture [50,53,54]. The analysis of local solutions used by rural inhabitants to overcome crises, no matter what form they take, is very important to us.

According to Nistor et al. [55], the traditional Romanian diet is based on meat and pork. Pork is the preferred meat of Romanian consumers and comprises about half of the total national meat consumption. Moreover, despite some worldwide and regional dietary trends that emphasize healthier food choices, particularly in developed, high-income countries in the EU [8], annual meat consumption in Romania has increased from 54.4 kg to 76.7 kg per person, with a particular preference for pork consumption. As meat production decreased, pork imports increased in Romania, doubling in the first nine months after joining the EU and covering about 70% of the domestic demand of local industry processors [55].

This situation can be explained by the fact that pig breeding in Romania mainly occurs on private, household family farms, further emphasizing the preservation of a traditional occupation. Pig breeding and their slaughter before Christmas has been celebrated for hundreds of years in Romania and numerous symbols and laic rituals were added to this event that occurs on 20 December on Ignat day around the time of the winter solstice. The word Ignat derives from the ancient Latin word ignatus and was initially used to refer to a solar deity. Over time, the event also took on Christian connotations as it took the name and date of the celebration of the orthodox saint Ignatie Teofanul. The sacrifice of the household pig is, therefore, done on the day of St Ignat around the time of the winter solstice when the sun is at its lowest point in the sky by slaughtering the animal

immediately after the sun rises and then cleaning it with fire and water. Therefore, this sacrifice is made to celebrate the rebirth of the seasons, coinciding with the rekindling of the sun, and it is also a purification ritual performed with the help of fire (ignis = fire). In ancient times, this ritual was performed to chase away the bad spirits associated with the ending of the dark and cold winter season and ensure the health and prosperity of the family, as well as good crops and fertile land in the new year and the new spring to come [27,56]. These traditions have been rigorously maintained, especially in rural areas that nowadays advertise authentic cultural tourism, and have influenced the local cuisine and dietary behavior of many Romanian people.

Moldovan [57] stated that over 79% of pig livestock in Romania was raised on small individual farms, with pig farms comprising an impressive number of about 1.7 million nationally, with an average of 2.8 heads per farm in 2007. The importance of swine breeding in Romanian communities and its effects on the pork meat sector was emphasized in the European Parliamentary Research Service report, which stated that more than half of pig farms in the EU are located in Romania, which has one of the most polarized pig industries in Europe [10]. According to this report, approximately 99% of Romanian pig farms have less than 10 pigs, yet these farms account for half of the country's pig population. This emphasizes the importance of this traditional occupation, which is simultaneously a socio-economic occupation and a cultural feature of the local rural landscape. The above figures support the fact that pork is a traditional product consumed by the general Romanian population.

## 3. Research Methodology

### 3.1. Case Study

The analysis of swine breeding was performed for Vâlcea county which is located in the central southwestern part of Romania in Oltenia province (Figure 1).

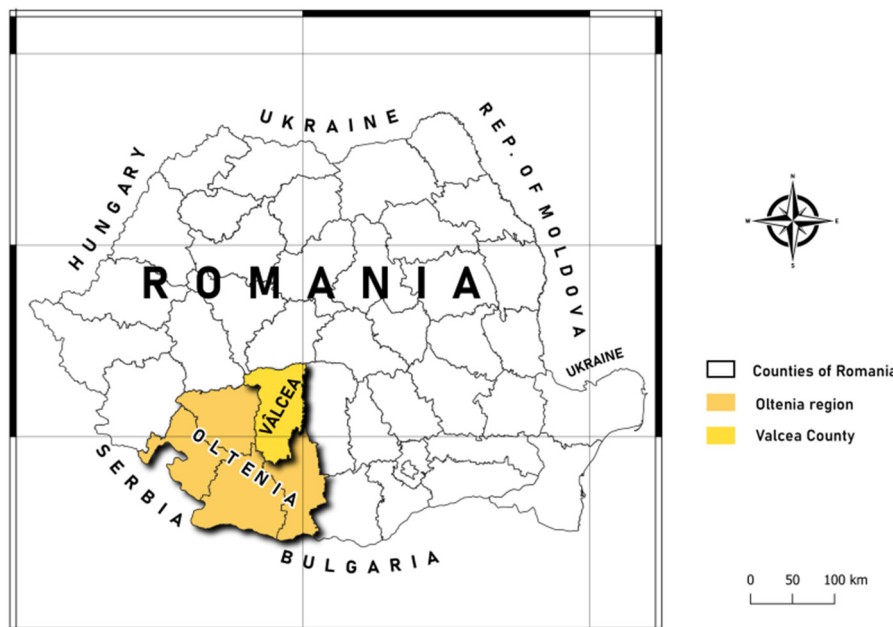

**Figure 1.** Study area—Vâlcea county, southwest development region, Oltenia Province, Romania.

Particularly in Vâlcea county, pig breeding has a long history and officially dates back to 1388, according to the *aşa cum este menţionat într-un* document sent from the King of Wallachia, Mircea the Elder (Mircea cel Bătrân), to the Cozia monastery [24]. Although pig farming spread uniformly across the entire Romanian territory, southern regions have been found to contain greater numbers of pigs and higher pork production due to more favorable factors, in particular, cereal production [58,59], an aspect well demonstrated at a

microregional level by our case study. The main food source for pig farming in Romania is cereals, which are advantageous in regions with lower altitudes and a relief of plains and plateaus compared to mountainous areas. Therefore, southern settlements in Vâlcea county have a higher number of pigs per farm due to the agricultural advantages brought by the lower relief forms, which are more suitable for crop cultivation. According to official statistics, the average rural household in Vâlcea county has 2.85 ha of cultivated land, with about 50% of it used for corn and wheat cultivation. The crop yield is variable, with about 2800–3200 kg/ha of corn and 2000–3000 kg/ha of wheat, which allows a sufficient food surplus to raise a minimum of one pig per household.

Natural and socio-economic factors, as well as cultural and behavioral factors, as explained above, have influenced a complex situation in which the number of entrepreneurial initiatives for pig breeding in Vâlcea county (households and farms raising 2–4 pigs and 5–10 pigs) has increased, along with the proportion of families raising pigs in the last two decades (Figure 2, Table 1).

**Table 1.** Variations in the swine-raising index according to the number of pig-raising households and types of farms in Vâlcea county.

| | No. of Households (Families) Raising Pigs | | Swine-Raising Index $I_{cr}$ = Pig No./Total No. of Households (Families) in the County (Average * and Farm Type **) | No. of Rural Settlements with over 50% of Households (Families) Raising Pigs within the Total *** | |
|---|---|---|---|---|---|
| No. of households 1990 76,844 | 1 pig/household | 15,668 | 0.20 | | |
| | 2–4 pigs/household | 11,207 (22,414–44,828) | 0.14 (0.29–0.58) | ↘ | 34 |
| | 5–10 pigs/household | 6953 (34,765–69,530) | 0.09 (0.45–0.90) | | |
| | ≥11 pigs/household | 5207 (≥57,277) | 0.06 (≥0.74) | | |
| No. of households 2000 71,960 | 1 pig/household | 10,542 | 0.14 | | |
| | 2–4 pigs/household | 8645 (17,290–34,580) | 0.12 (0.24–0.48) | ↘ | 13 |
| | 5–10 pigs/household | 3668 (18,340–36,680) | 0.05 (0.25–0.50) | | |
| | ≥11 pigs/household | 2614 (≥28,754) | 0.03 (≥0.39) | | |
| No. of households 2010 66,050 | 1 pig/household | 11,099 | 0.16 | | |
| | 2–4 pigs/household | 9718 (19,436–38,872) | 0.14 (0.29–0.58) | ↗ | 17 |
| | 5–10 pigs/household | 4895 (24,475–48,950) | 0.07 (0.37–0.74) | | |
| | ≥11 pigs/household | 2834 (≥31,174) | 0.04 (≥0.47) | | |
| No. of households 2022 65,930 | 1 pig/household | 9761 | 0.14 | | |
| | 2–4 pigs/household | 10,480 (20,480–41,960) | 0.16 (0.31–0.63) | ↗ | 33 |
| | 5–10 pigs/household | 5742 (28,710–57,420) | 0.08 (0.43–0.87) | | |
| | ≥11 pigs/household | 3272 (≥35,992) | 0.04 (≥0.54) | | |

Source: Computed by authors. (* the average values of the swine-raising index refers to the ratio between the total number of households (families) raising pigs and the total number of households (families) in the county; ** the values of the index according to farm type refers to the ratio between the possible number of pigs raised according to the farm size and the total number of households (families) in the county; *** the total number of rural settlements in Vâlcea county is 78; ↘ decreasing trend; ↗ increasing trend).

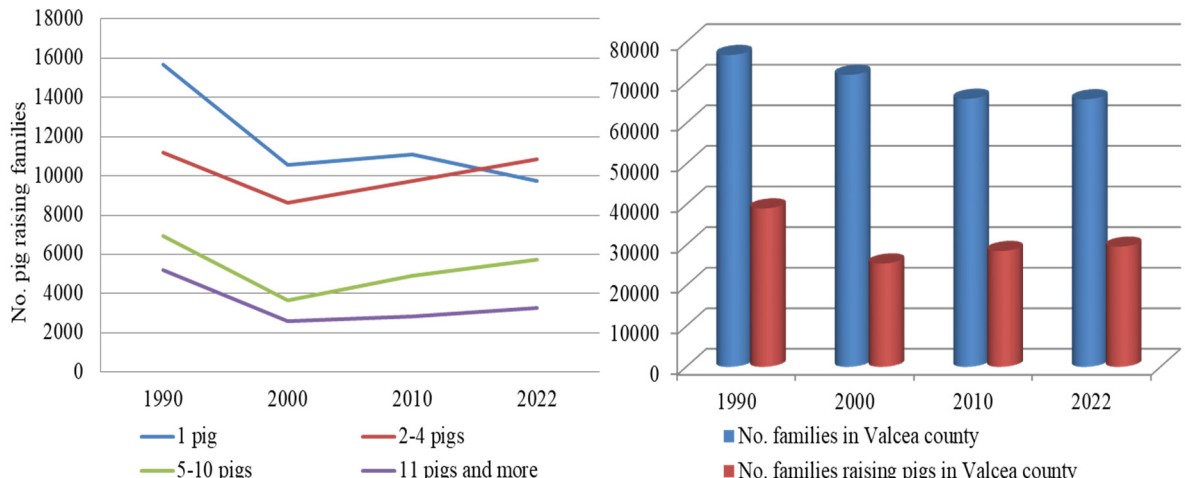

**Figure 2.** Evolution of pig farms andevolution of the number of families (households) raising pigs in Vâlcea county.

### 3.2. Research Methods

Our research questions and approach, which considers the territorial and evolutionary dimensions) of the phenomenon of pig farming in Vâlcea county, led us to adopt a mixed-method perspective that combines quantitative statistical and mapping software with qualitative techniques such as semi-structured interviews and aims to understand the motivations for small-scale swine farming in rural settlements in Vâlcea county. This mixed methodology, together with multi-method research [60], was needed to ameliorate the validity and reliability of the results, as neither statistics nor qualitative interview surveys alone could explain the complexities of this socio-economic phenomenon. In order to further extrapolate these possible behavioral changes to the entire Romanian territory, the data were analyzed at four distinctive time points in the post-socialist evolution, namely the post-revolution moment (1990), EU pre-accession (2000), the Romanian economic crisis (2010), and the contemporary period marked by the sanitary crisis and the military conflict in Ukraine (2022). Demographic elements, such as population aging and migration overseas and from villages to cities within Romania, overlapped with these major economic and social events in the four sequential time periods in our illustrative case study.

The first method used that aims to illustrate the spatial distribution and concentration of swine farming in the rural settlements in Vâlcea County for the four reference years based on existing demographic and economic resources was a cluster analysis, which has been used by other studies that focused on farming patterns in Romania [9]. The cluster analysis started with the data on households involved in pig farming. Households were classified into those with 1 pig, 2–4 pigs, 5–10 pigs, and over 11 pigs. A supplementary category with households with 0 pigs was created in order to include the total number of households and obtain the proportion of households involved in swine farming.

To expand this study, the authors consulted both scientific literature and existing analyses of the socio-economic context of Romania in the post-socialist period.

In order to explain the rules of pig farming in Romania, the study references current legislation. In this case, Order 280/2019 of the Minister of Agriculture specifies the existing rules for pig farms that do not require public health official approval [61]:

- animals raised for food: subsistence pig holding—agricultural holding with a maximum of 5 pigs;
- animals raised for commercial purposes: small pig holding—a farm with a total of between 6 and 65 pigs.

The time needed to perform the fieldwork to obtain reliable statistics (using the city halls of the rural settlements in the county as data sources), along with the ample

interview-based surveys that were conducted in July 2022, led us to select Vâlcea county as a representative sample of Romania, as it has one of the most diverse relief forms and balanced proportions in the country, making it an ideal case-study location. Unlike other large-scale studies that focus on a multitude of backgrounds and where case-study results are limited to a specific territory [62], our research has the advantage that its results that can be extrapolated to the entire territory. This is because the studied territory is representative and perfectly illustrates the geographic, socio-economic, and legislative background of a NUTS 2 administrative unit in Romania. The case-study strategy is particularly useful in social sciences for collecting and analyzing empirical evidence about a topic and focusing on descriptive and explanatory issues for the analyzed phenomenon [63], as in the case of our research. According to the research strategies matrix described by Yin p. 5, Ref. [63], we considered our research topic and aimed to answer questions such as how many farms are involved in swine breeding and for what purposes, how swine farming is concentrated, and what reasons motivate farmers in post-socialist Romania to raise pigs. Based on this, we determined that both survey and case-study approaches would be appropriate research strategies to investigate this topic.

*3.3. Data Gathering and Processing*

The research questions, methodological framework, and research strategy constituted a threefold methodological approach that utilized data processing and analysis.

The application of statistical methods resulted in two clusters. We also included other relevant and officially available statistical variables such as population and land-use structure. One of the limitations we faced was the lack of official data characterizing Romanian agriculture, which prevented us from using more complex statistical methods and tests such as multi-stratified clusterization. Although there have been initiatives such as the National Structural Survey in Agriculture performed by the National Institute of Statistics in 2016 [64], which provide a cross-sectional study snapshot of the sector, they do not reflect its evolutive trends.

In order to obtain the clusters for each of the analyzed years, the first step was to measure the clustering tendency. This was calculated using the Hopkins command from the Hopkins R package and the values obtained were 0.9987 for 1990, 0.9327 for 2000, 0.9811 for 2010, and 0.9999 for 2022. All these values are well above 0.7, indicating the presence of clearly demarcated clusters.

The next step was to determine the optimal number of clusters. The analysis was performed using the WSS (within-cluster sum of squares), average silhouette, and gap statistic methods, which are available in the factoextra package. The results were then confirmed using the NbClust package, which uses 30 indicators to identify the optimal number of clusters. For all four time points, the number of clusters most often suggested was two.

Some methods recommended up to 16 clusters but we opted for 2 clusters for each time point as it was the most suggested solution.

For instance, for 2010, the suggested number of clusters was 3 but we opted for 2 clusters because 3 clusters were suggested by 9 of the identification methods, whereas 2 clusters were suggested by 8 of the methods. As this was not a significant difference, keeping the number of clusters at 2 was the most appropriate solution, as it allowed us to compare the clusters and their evolving trends in the four selected years.

Next, we identified the clusters themselves using the K-means clustering method and the Euclidean distance as the measure of distance. K-means clustering is a method of identifying clusters in which all values are randomly assigned to the specified number of clusters, in this case, 2.

Next, the center of each cluster was calculated and then the distance from each value to this center. If a value was closer to the center of another cluster than to the center of its own cluster, that value was assigned to the cluster to which it was closer. After all these

reallocations were made, the process was repeated and a new average was calculated. The process stopped when there were no more reallocated values.

A second methodological step utilized descriptive quantitative and mapping techniques for the four analyzed years. The statistical data used for this method were collected by the first author of this paper at a rural locality level through fieldwork. Data on the number of pigs raised in rural households, the proportion of pig-farming households, and the total number of households in each commune were gathered for the categories of farm ownership: 1 pig, 2 to 4 pigs, 5 to 10 pigs, and more than 11 pigs.. This method allowed us to analyze the numerical oscillations for the specified period in rural areas of Vâlcea County and highlight the areas mainly involved in swine breeding in the analyzed territory. The evolution of each category of farm based on the livestock dimensions in the four years was analyzed.

The third method employed in the study was the survey method, which is essential to the understanding of this topic [64,65]. To complement the information acquired using quantitative and statistical methods, interviews were used to determine the motivations for pig farming. A total of 126 interviews were conducted in various settlements, aiming for a balanced number of interviews in different types of settlements according to socio-economic, demographic, and geographic criteria. Many responses were similar and we analyzed the responses and their frequencies in order to extract relevant examples that reiterated the main reasons for swine farming in Vâlcea county.

The reasons for raising pigs in rural households in Vâlcea county are varied due to different familial situations and have been influenced by previous socialist or current free market economic laws. Direct dialogue with farmers, combined with the fieldwork, helped in the interpretations of the graphical representations, particularly the data related to the number of pigs raised in rural households in each analyzed year. The resulting information supported the main findings and conclusions of the study.

## 4. Results

### 4.1. Demographic and Economic Factors Stimulating Spatial Distribution and Concentration of Swine Farming in Rural Settlements in Vâlcea County

The main results of the study are presented according to the aims, objectives, and methodological framework.

Based on the cluster analysis, as explained above, two clusters of rural settlements were predominantly formed (Figures 3–6).

The localities that fell into the two types of clusters were usually the same, with a few exceptions. In all the years investigated (1990, 2000, 2010, 2022), the localities with higher concentrations of pigs were those in cluster 2, (Figures 3–6). This trend was observed for all categories of households: those with 1 pig, with 2–4 pigs, with 5–10 pigs, and with 11 pigs or more.

Another aspect that should be noted is that the average population size of the settlements in cluster 2 was lower than that of the settlements in cluster 1. At the same time, the settlements in cluster 2 had a higher share of agricultural land in their land-use structure and a higher percentage of land dedicated to cultivation and pasture than those in cluster 1.

An interesting aspect of this analysis is the evolving clusters, where settlements in cluster 1 migrated toward cluster 2. Factors that contributed to the cluster changes between 2010 and 2022 included a rapidly aging population and population migration, which accentuated the share of agricultural land for the remaining population in rural areas. Cereal crops that favor larger pig farms were located in the southern parts of the county, which is dominated by lower relief forms along the Olt Valley rivers.

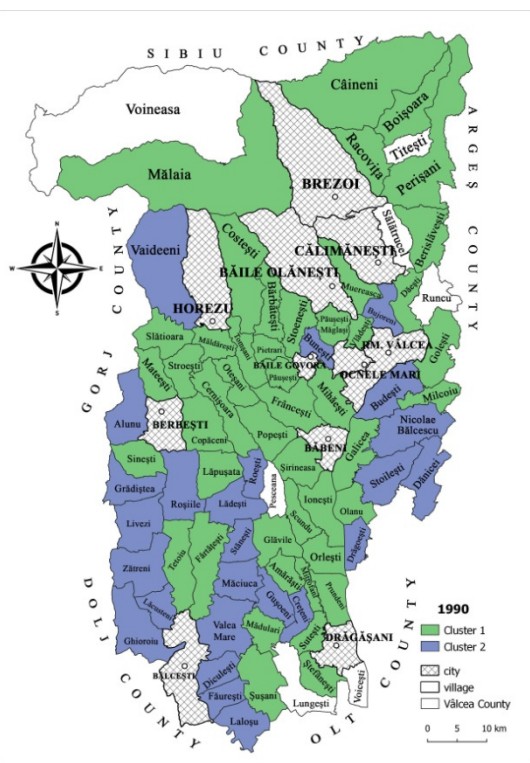

**Figure 3.** Clusters of rural settlements with pig-farming households in Vâlcea county in 1990. Source: elaborated by authors.

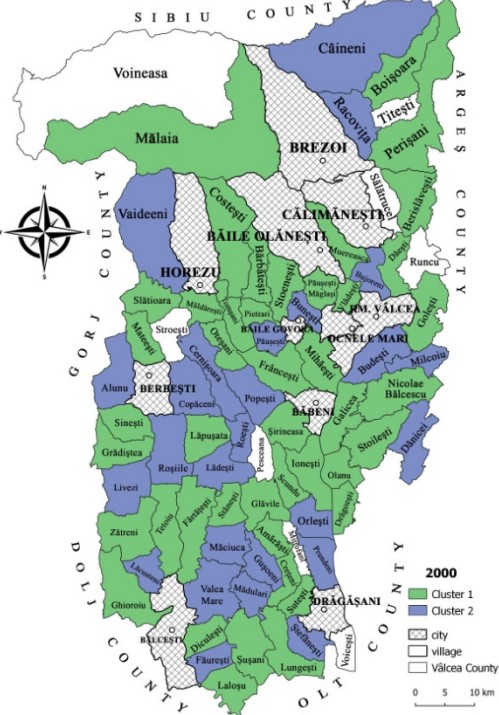

**Figure 4.** Clusters of rural settlements with pig-farming households in Vâlcea county in 2000. Source: elaborated by authors.

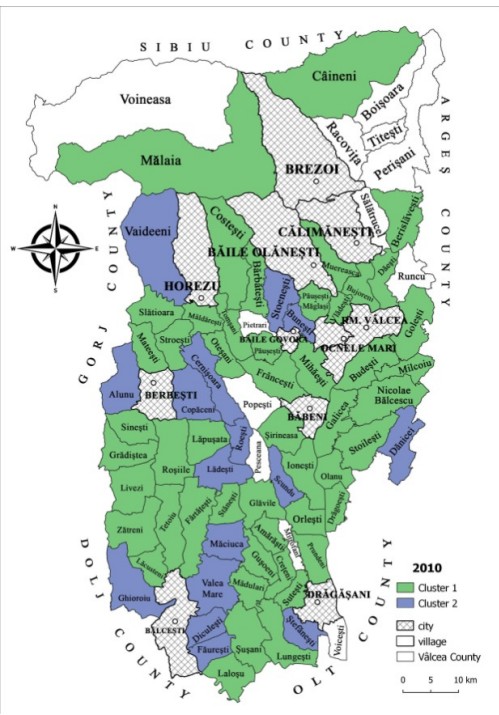

**Figure 5.** Clusters of rural settlements with pig-farming households in Vâlcea county in 2010. Source: elaborated by authors.

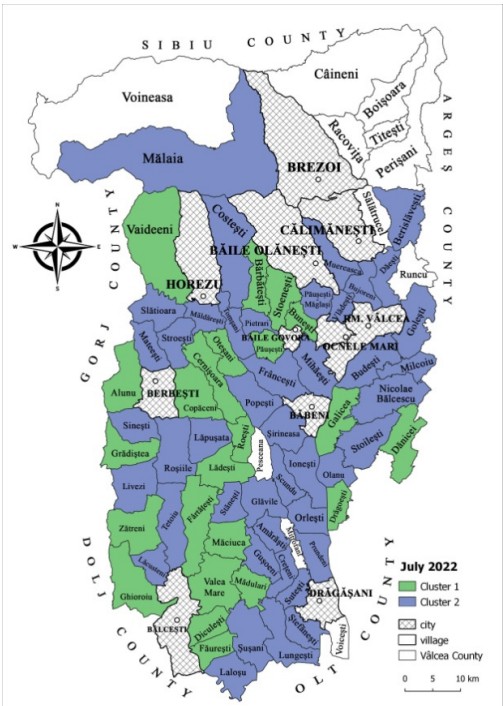

**Figure 6.** Clusters of rural settlements with pig-farming households in Vâlcea county in July 2022. Source: elaborated by authors.

Mapping techniques also helped to illustrate the differences in the spatial distribution of swine farming in the rural settlements of Vâlcea county. The maps display important differences in the distribution of swine farms at a local level and from one year to another (Figures 7–10). From a statistical point of view, the distribution suggests a growing trend toward commercialization in certain villages (settlements including households with more

than 5 pigs) but the general tradition of swine breeding was maintained and even reinforced in the studied region. This characteristic addresses the need for a diverse and efficient rural economy.

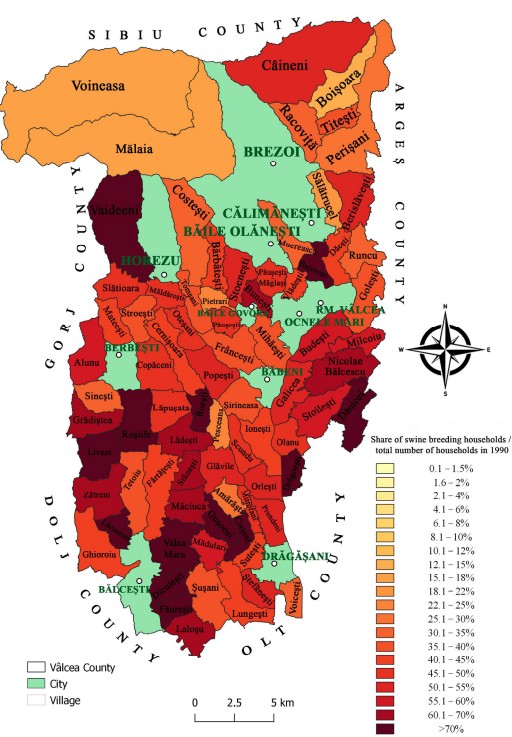

**Figure 7.** Share of swine-breeding households/total number of households in rural settlements in Vâlcea county in 1990. Source: elaborated by authors.

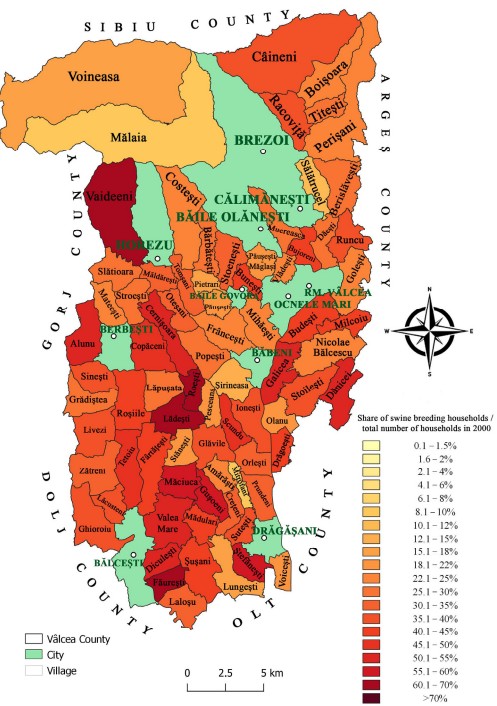

**Figure 8.** Share of swine-breeding households/total number of households in rural settlements in Vâlcea county in 2000. Source: elaborated by authors.

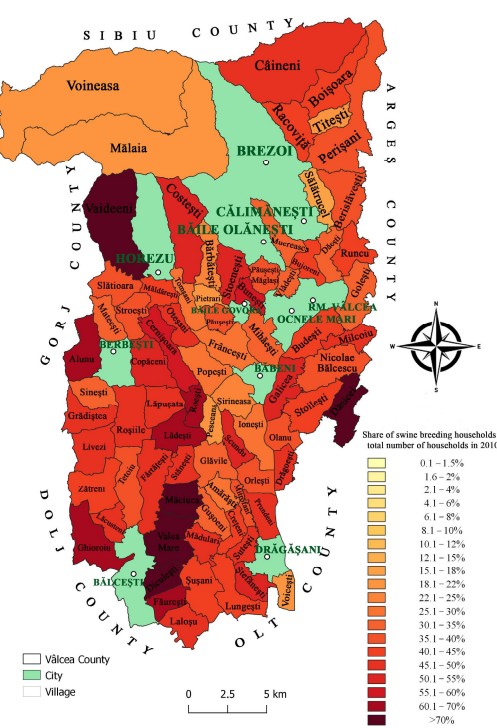

**Figure 9.** Share of swine-breeding households/total number of households in rural settlements in Vâlcea county in 2010. Source: elaborated by authors.

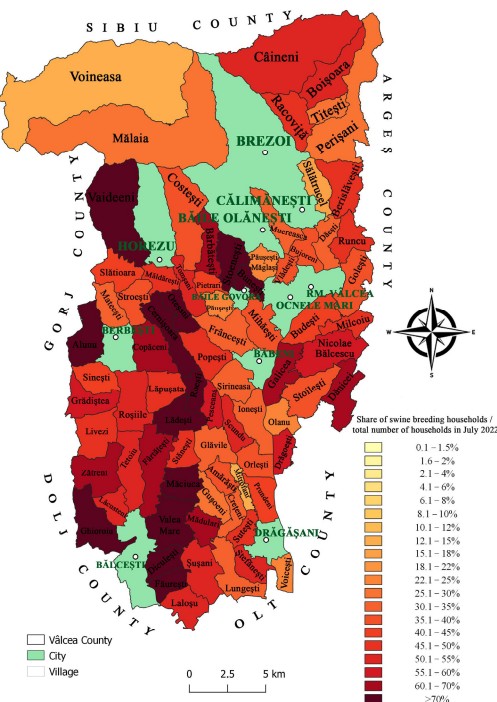

**Figure 10.** Share of swine-breeding households/total number of households in rural settlements in Vâlcea county in July 2022. Source: elaborated by authors.

The generation of organic food and ecological products increases the quality of life and the commercialization of fresh pork and livestock brings economic advantages and supplements household incomes. These activities emphasize respect for the consumer while also respecting the environment [57]. In the year 1990 (Figure 7), the influence of the communist economic system was still present. However, the year 1989 marked a turning

point for the Romanian economy and politics but human domestic activity, especially in rural areas, maintained an inertia that had accumulated over almost half a century. Consequently, quite a high proportion of peasant households owned at least one pig. This type of household dominated the rural economy of Vâlcea county for decades.

By the year 2000 (Figure 8), pig farming was affected by the appearance of chain stores (of Western European origin), which led to a proportion of peasants in rural areas giving up the traditional occupation of swine breeding that had previously supplied their families with meat products.

Economic factors also influenced the concentration of swine breeding at a local level in rural areas in Vâlcea county. The relatively large number of cities in the region influenced swine breeding in the neighboring villages. The need for cheap and organic fresh pork led to the intensification of swine farming in rural settlements in the year 2010 (Figure 9). After February 2020 (when the first case of COVID-19 was reported in Romania), the sanitary crisis and its associated socio-economic impact influenced the further development of rural swine-breeding activities. The pandemic limited travel and, therefore, the transportation of food over long distances. It also led to unemployment and a loss of revenue for part of the population. Swine breeding was reoriented toward rural areas that were less affected by the sanitary crisis and had cheaper residences, thereby providing the means to produce food and attain self-sufficiency (Figure 10). However, each rural settlement has its own social, demographic, and economic particularities.

The behavior of the population regarding swine breeding in their households was influenced by the regional economic profile, which evolved in the post-communist period under the influence of various factors (Figure 11).

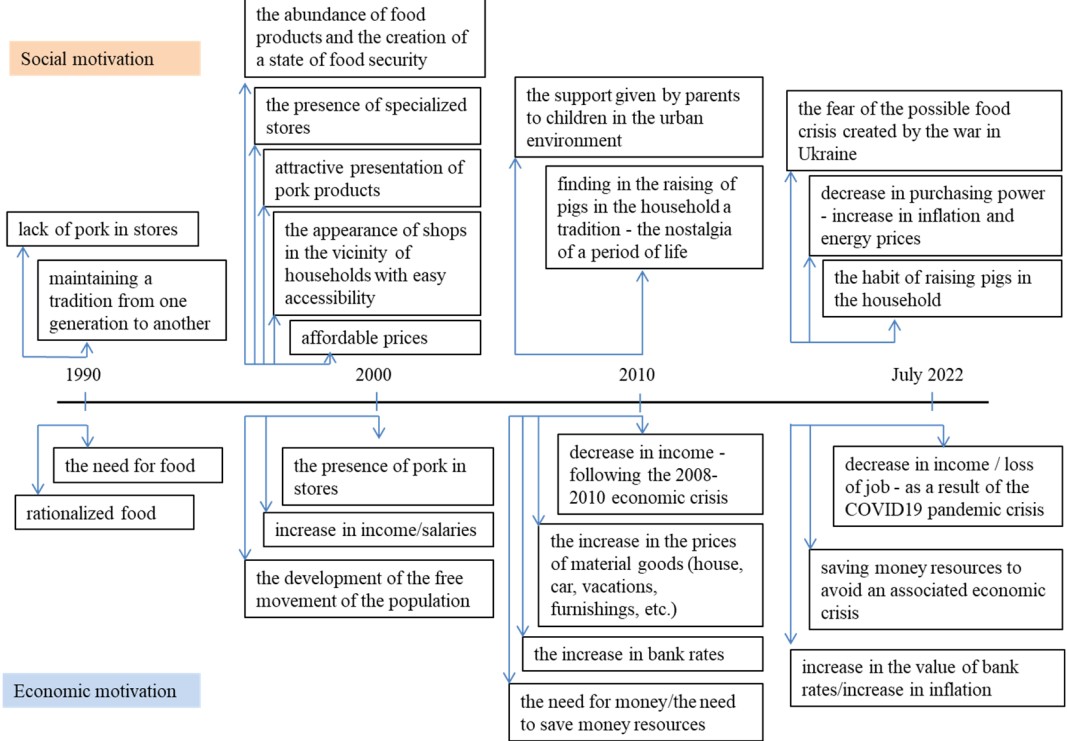

**Figure 11.** The motivations offered by rural inhabitants for swine breeding—diachronic perspectives. Source: elaborated by authors.

Cereal crops required considerable land, which, in turn, led to a more intensive pig-breeding approach among households and farms. This explains the higher proportion of swine-breeding households in the southern parts of the county in 1990.

According to statistics obtained from the local administration of each commune, over 50% of households in 79 rural settlements had at least one pig, whereas 70% of households in 12 settlements raised pigs.

In the year 2000, the context changed. Two distinct situations contributed to an aging population in the rural areas of Vâlcea county: (a) the migration of many working-age people away from the area; and (b) the permanent migration of young people for education and work abroad. As a result, the farmers in rural settlements became pork providers to supply the nearby cities, which boosted their revenues while also providing meat for their own consumption.

The economic crisis of 2008–2010 caused a shift in population from urban environments to their rural areas of origin, where they had previously not sold properties or taken advantage of the lower house acquisition costs compared to cities. The diminution of incomes, the economic recession, and the increase in real estate prices in the cities led to "an exodus" of people over 55 from urban areas, who moved to the neighboring rural settlements or their areas of origin. The inflation rate, the diminution of incomes, and the lack of work opportunities were the main causes of urban emigrants returning to rural areas (Figures 8 and 9). Therefore, with the newcomers in the rural areas, there was an increased need to secure food, which multiplied the number of households with at least 1 pig. Some farms also expanded their swine-breeding activities, by deciding to raise more than 5 pigs.

The year 2022 was unique due to the sanitary crisis, which imposed restrictions at both the international and national levels and influenced the return of people to rural areas. The phenomenon of returning to the area of origin and reinhabiting real estate and agricultural land was accelerated by the regional conflict in Ukraine. The fear of a possible food crisis generated by the military conflict caused many inhabitants to shift toward family farming, including swine breeding—an activity with a long tradition in Vâlcea county.

The COVID-19 pandemic once again transformed the local economy. It influenced the return of many people abroad to their areas of origin, who then invested in small farms (those with more than 11 pigs) (Figure 10). This type of business covered both the family's consumption needs and ensured part of the household revenue. Another factor that supported this business was represented by young people who worked remotely via the Internet, as well as people who had lost their jobs and returned to rural settlements to revitalize their socio-economic environment. The economic recession, high energy prices, and the increases in inflation and lending rates caused many urban inhabitants to reconsider their countryside homes as a source of food and a way to reduce their expenses.

The war in Ukraine and the fear of a military conflict were supplementary reasons for the population shifting toward rural areas and utilizing swine breeding as a traditional way to obtain food. By looking at the number of pigs per household (1 pig, 2–4 pigs, 5–10 pigs, 11 pigs or more), it is possible to distinguish several categories of farms based on their orientation toward traditional or economic benefits. The areas dominated by the first two categories of households that raised a low number of swine were mainly tradition-oriented. Households raising one pig were oriented more toward self-sufficiency and the preservation of traditional activities. In this type of household, and often for households raising between two and four pigs, there was no option for pig commercialization. Generally, farms raising up to four pigs were households where two or three families of different generations lived (e.g., parents and children). Unstable financial circumstances and the tradition of several families living in the same household are often encountered in the rural environment of Oltenia region, Romania, to which Vâlcea county belongs. Consequently, the commercialization rate for this second type of farm is low (below 15%).

*4.2. Tradition and Its Evolving Nuances for Swine Breeding in Rural Settlements in Vâlcea County*

4.2.1. Inherited Elements from the Communist Period

The communist era was characterized by a centralized economy that imposed rationed food consumption on the population, especially meat and meat products. According to

these "rations", a person was entitled to 300 g of bread per day and monthly rations of 500 g of cheese, 10 eggs, 500 g of pork or beef, 1 kg of poultry, 100 g of butter, 1 kg of sugar, 1 L of oil, and 1 kg of flour. In order to mask the food crisis, a program of "rationed nutrition" was invented, declaring it unhealthy for an adult to consume more than 3000 calories per day [66]. The ration of 500 g of pork/person/month could be compensated for only by raising pigs in private households in the countryside. Reliable meat sources were parents and siblings who lived in the countryside. In the year 1990, numerous households raised one pig in Vâlcea county's rural settlements to ensure self-sufficiency in fresh meat and meat products (Figure 12). This reflects an important tradition rooted in the local population's mentality that emphasizes the importance of satisfying their need for fresh meat. Households with a larger number of people and/or families, and, therefore, with a greater need for meat products, raised more than one pig/household, typically ranging from two to four pigs (Figure 13).

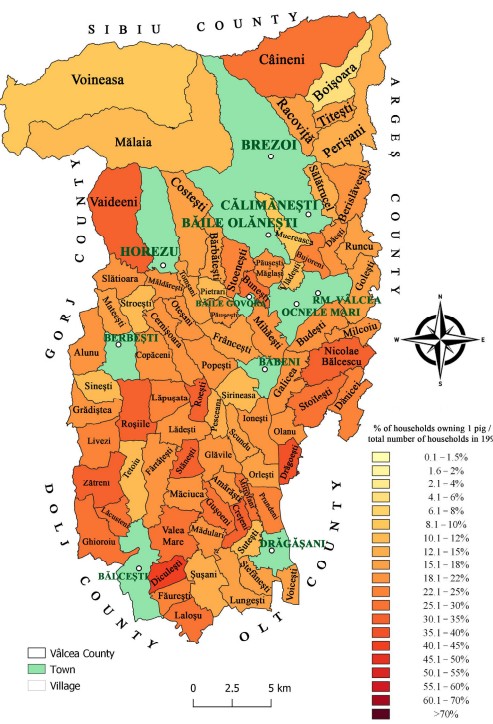

**Figure 12.** Share of households owning 1 pig in the total number of households in the rural settlements in Vâlcea county in 1990. Source: elaborated by authors.

The preservation of local tradition was also highlighted by residents mentioning the importance of preserving pig breeding in their household.

*"My grandparents raised pigs in their courtyard, and my parents also raised pigs. How can I do otherwise? It is normal to raise pigs for me and my children. Why should I buy pork? I have everything I need to raise at least one!"* (Mihaela, 39 years old, Diculeşti).

*"I am quite old, but I have a pig in the yard. I talk to him; I have a reason to get up in the morning. I have to feed it. I have a purpose in this world. Now I feed it. Then it is its turn to feed me all year long. Each year I will raise one pig as long as I live!"* (Nicolae, 74 years old, Alunu).

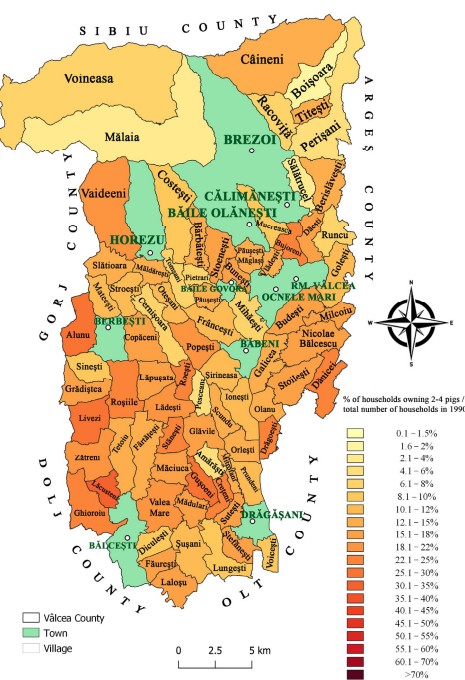

**Figure 13.** Share of households owning 2–4 pigs in the total number of households in the rural settlements in Vâlcea county in 1990. Source: elaborated by authors.

4.2.2. Ongoing Tradition in Vâlcea Rural Settlements

After the fall of the communist regime, market economy relations were quickly established. Regardless of their living environment, the population was attracted to new lifestyles that were previously unavailable to them in the previous period. Consumption choices should focus on three elements: health, sustainability, and convenience [6].

Trade liberalization, including meat and pork products, caused Romanian urban consumers to purchase from markets and not rural households. At the same time, there were changes in the behavior of the rural population as inhabitants of villages also bought many products from stores and supermarkets and gave up swine breeding. Ten years is considered optimal for stabilizing consumer preferences and collective behavior in a geographical area.

For the year 2000, a reduction in the number of pigs raised was recorded in most rural settlements in Vâlcea county (Figures 14–17). The causes of these decreases were:

- the movement of working-age young people to nearby urban centers;
- the population's preference for purchasing various commercialized products, including pork, from supermarkets;
- the diminution of pork consumption;
- the substantial reduction and even elimination of secondary products generated by pig breeding (e.g., homemade soap from pork fat).

Despite these new trends in consumption that reduced the demand for pork, the local tradition of swine breeding persisted in the year 2000 in Vâlcea county (Figures 14 and 15).

In response to the question of the possible abandonment of pig breeding, locals clearly expressed a diminished interest in this traditional activity because of trade liberalization and development:

*"The store is near my house. It is not worth raising pigs as you can find everything you want at the store. I have elevated enough for me and for my children. Now they are gone, and I have no one to grow up pigs for!"* (Ioana, 68 years old, Ghioroiu);

*"I don't have a pig, and I don't need one. I can't eat pork because I am old and sick. If I want to eat pork, I take a kg of meat, and that is enough. Now you can find it; it is not like during the time of communists."* (Victor, 82 years old, Roşiile).

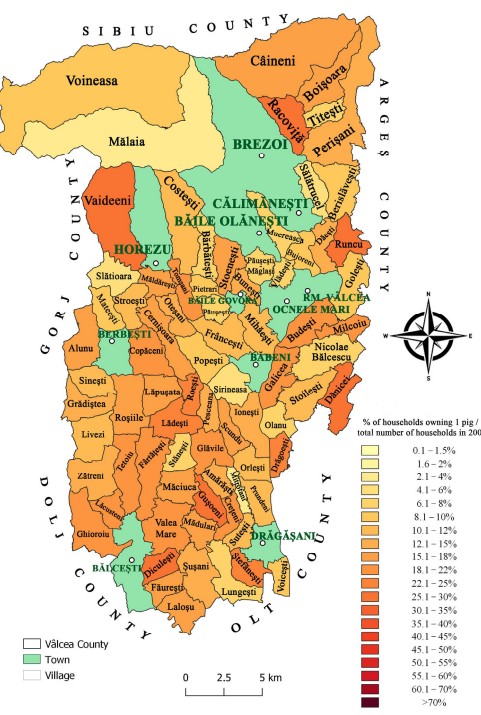

**Figure 14.** Share of households owning 1 pig out of the total number of households in rural settlements in Vâlcea county in 2000. Source: elaborated by authors.

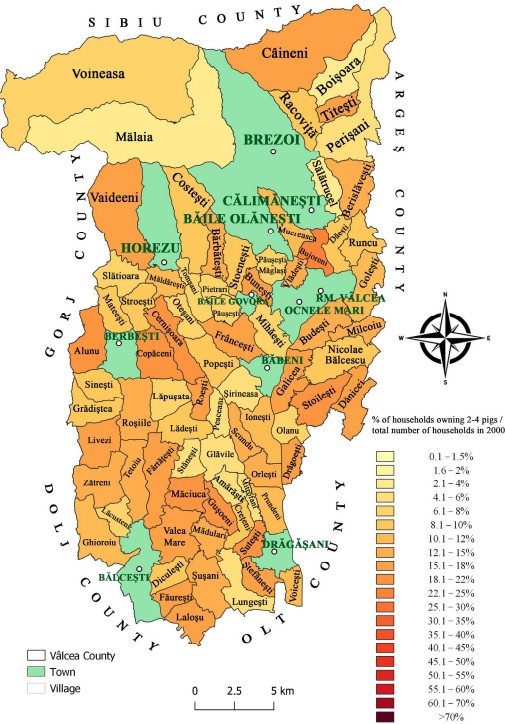

**Figure 15.** Share of households owning 2–4 pigs out of the total number of households in rural settlements in Vâlcea county in 2000. Source: elaborated by authors.

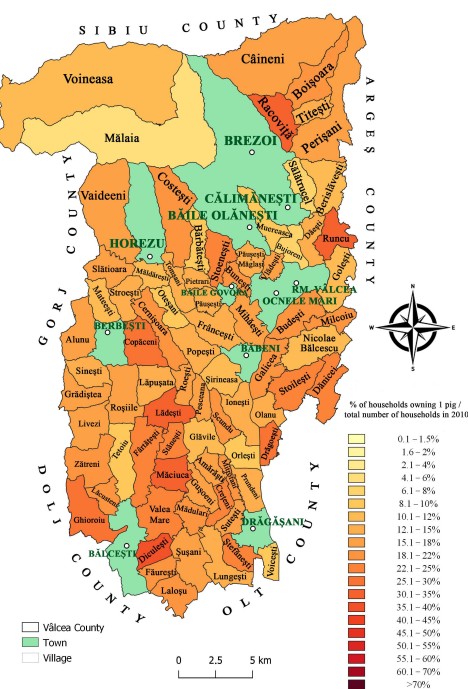

**Figure 16.** Share of households owning 1 pig out of the total number of households in rural settlements in Vâlcea county in 2010. Source: elaborated by authors.

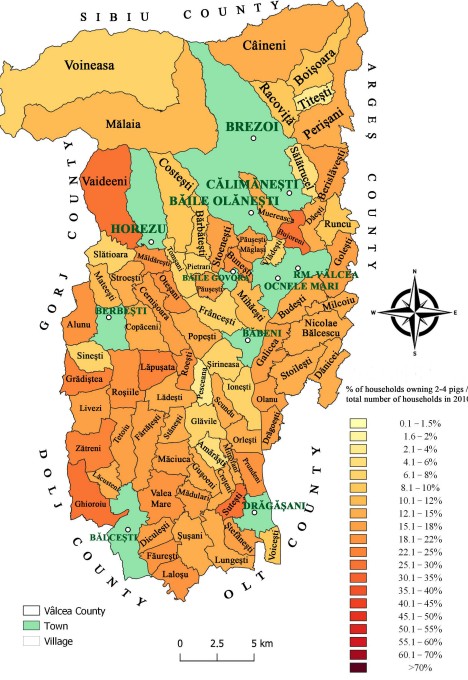

**Figure 17.** Share of households owning 2–4 pigs out of the total number of households in rural settlements in Vâlcea county in 2010. Source: elaborated by authors.

Despite the reduction in swine-breeding activities, other elements such as the economic recession and financial crisis affected a large part of the population. However, their personal situation was the reason that most households (more than 50%) raised at least one pig every year. This is why in 2010, regardless of family demographics, households raising one pig or two–four pigs were still very important in rural settlements in Vâlcea county (Figures 16 and 17). This reflects more than just tradition, as it also indicates an aspect of local culture.



In 2022, the most recent year of our diachronic analysis, the COVID-19 pandemic was a supplementary stressor, with lockdowns causing limited access to urban stores for rural inhabitants and socio-economic losses such as unemployment and decreased incomes. These stressors further reinforced the tradition of swine breeding in the studied area. As a result, people revived former occupations that had previously helped them to survive restrictive situations (Figures 18 and 19).

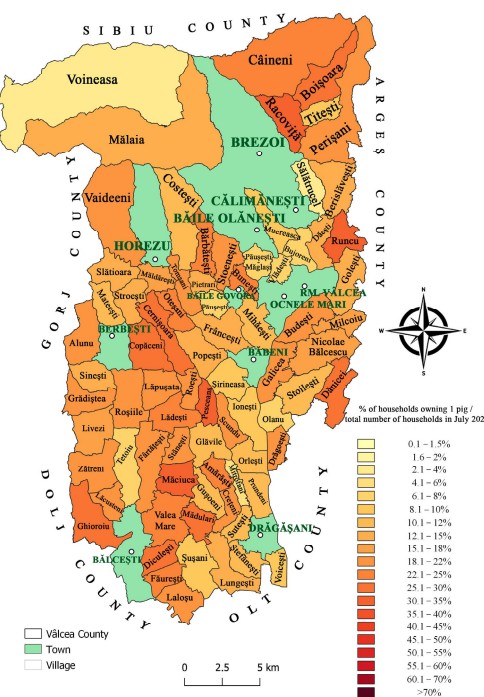

**Figure 18.** Share of households owning 1 pig out of the total number of households in rural settlements in Vâlcea county in July 2022. Source: elaborated by authors.

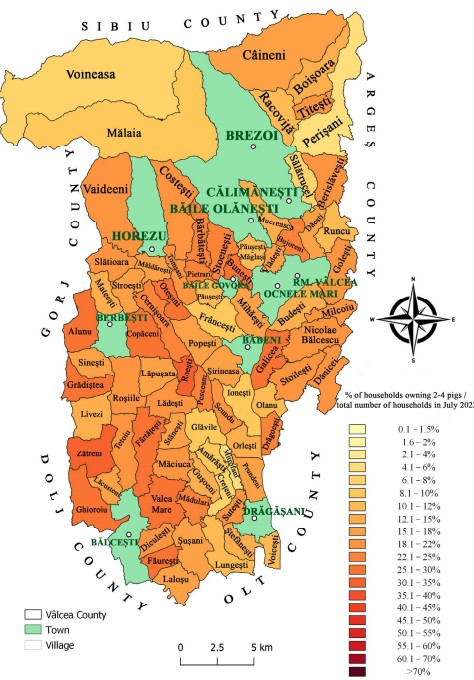

**Figure 19.** Share of households owning 2–4 pigs out of the total number of households in rural settlements in Vâlcea county in July 2022. Source: elaborated by authors.

### 4.3. Economic Necessity of Swine Breeding in the Post-Communist Period in Rural Settlements in Vâlcea County

4.3.1. Commercialization and Supplemental Revenue

Because of the pre-existing political and economic context in 1990, very few rural households in Vâlcea county had 5–10 pigs. This situation was influenced by both the difficulties of providing food for animals and the difficulties of selling and valuing livestock (Figure 20). Some households in communes located near cities (e.g., Bujoreni, Păuşeşti) or with a dominant agricultural identity (Vaideeni) managed to own between 5 and 10 pigs in 1990.

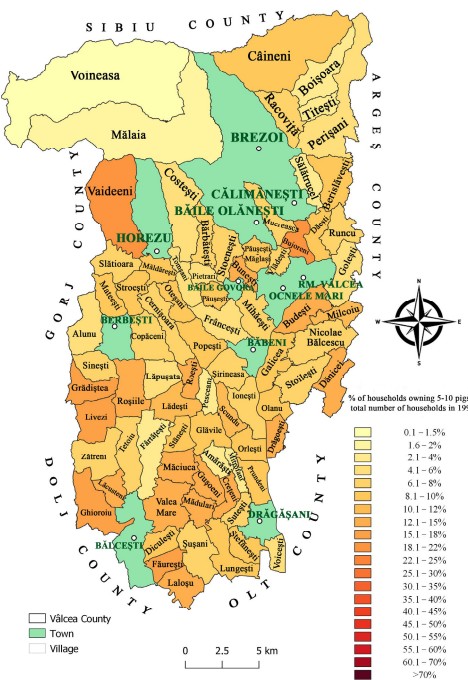

**Figure 20.** Share of households owning 5–10 pigs out of the total number of households in rural settlements in Vâlcea county in 1990. Source: elaborated by authors.

Households that owned more than 11 pigs in 1990 were mainly located in the communes of Budeşti, located near the county's capital city, Râmnicu Vâlcea; Făureşti and Diculeşti, which were able to provide important food resources for animals as they were located in areas with cereal crops; and Ghioroiu and Valea Mare, which also contained cereal-intensive cultivation areas (Figure 21).

The disappearance of jobs combined with the economic decline was accentuated for certain industry sectors (mining, forestry, local manufacturing), and created the need for professional reorientation in certain communes and cities in Vâlcea county. Agriculture and swine breeding were two solutions, and farms with 5–10 pigs (Figure 22) or more than 11 pigs (Figure 23) sprung up in the late 1990s. In these cases, tradition and economic necessity coexisted, as pork surplus was sold after satisfying household needs.

For households with insufficient revenues, after the year 2000, farmers focused more on selling fresh meat and meat products. Numerous cultivated lands were restored and became private property, which encouraged farmers to increase their pig numbers. The motivation to increase the family revenue, as well as the favorable existence of large, arable areas, led to the dominant cultivation of cereals, which, in turn, encouraged the remaining rural population to raise a higher number of pigs. In the year 2000, households raising 5–10 pigs and more than 11 pigs were found in settlements such as Roeşti, Valea Mare, Făureşti, and Ştefăneşti, which constituted between 12% and 16% of farms of this size.

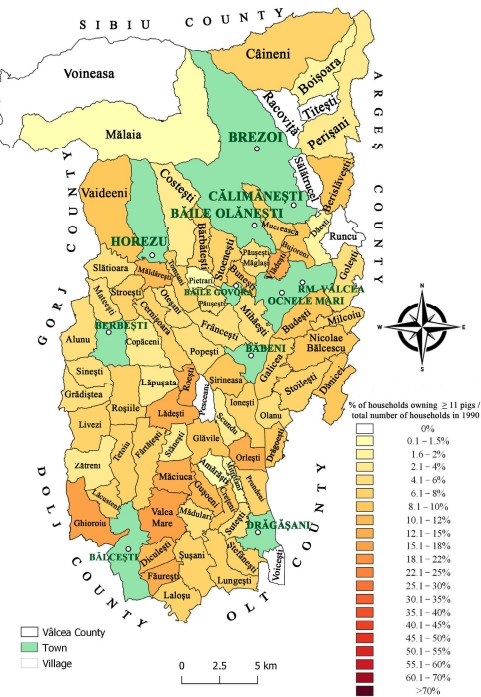

**Figure 21.** Share of households owning ≥11 pigs out of the total number of households in rural settlements in Vâlcea county in 1990. Source: elaborated by authors.

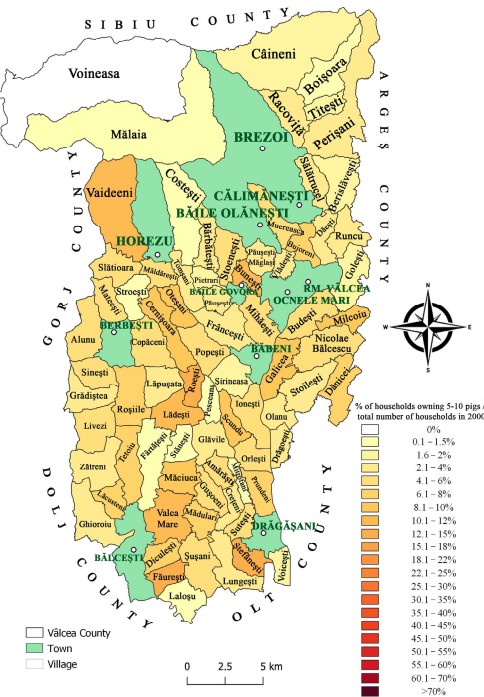

**Figure 22.** Share of households owning 5–10 pigs out of the total number of households in rural settlements in Vâlcea county in 2000. Source: elaborated by authors.

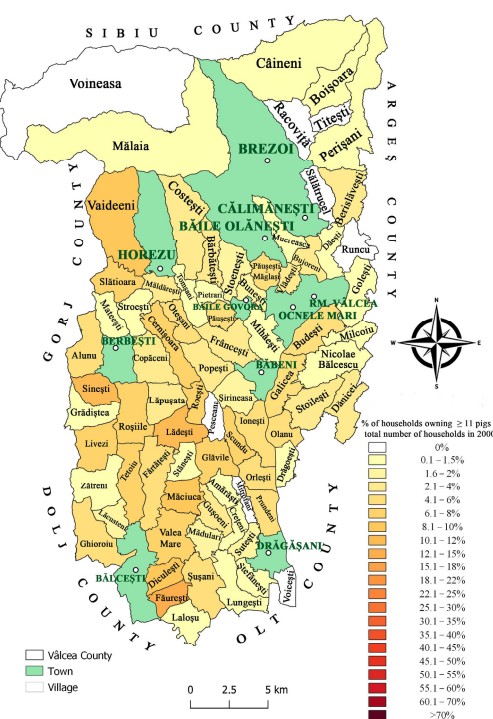

**Figure 23.** Share of households owning ≥11 pigs out of the total number of households in rural settlements in Vâlcea county in 2000. Source: elaborated by authors.

4.3.2. Relationship between the Economic Recession, Pandemic, Other Regional Crises, and Primary Food Products Such as Pork

The financial crisis that affected Europe in the summer of 2008 also affected Romania. After Romania entered the European Union, the quality of life in Romania became an important issue [67]. The recession forced the population to adopt protective measures, including increasing meat and pork production. According to statistics, over 21% of family expenses were allocated to food [55]. Awareness of savings on yearly food expenses, as well as the relatively small amount of resources needed to raise pigs in rural households, led to an increase in larger-sized households that raised 5–10 pigs or more than 11 pigs in 2010 (Figures 24 and 25).

In 2022, providing fresh meat for consumption and commercialization was particularly important in a regional context. As a result of the pandemic, of the military crisis in the region, inflation, and insufficient income, a large proportion of the population experienced financial problems.

A growing number of urban citizens began buying meat products from rural areas. This stimulated the economy for local producers and agricultural entrepreneurs. The number of households that offered pork meat and met sanitary-veterinary standards slightly increased compared to the previous periods analyzed in our study (Figures 26 and 27).

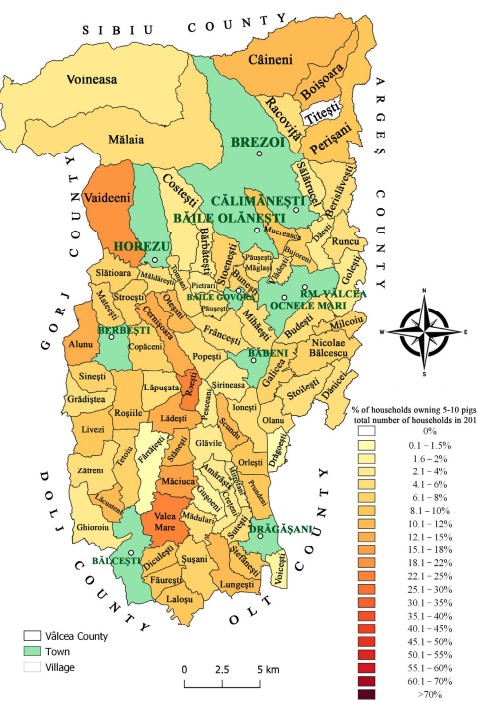

**Figure 24.** Share of households owning 5–10 pigs out of the total number of households in rural settlements in Vâlcea county in 2010. Source: elaborated by authors.

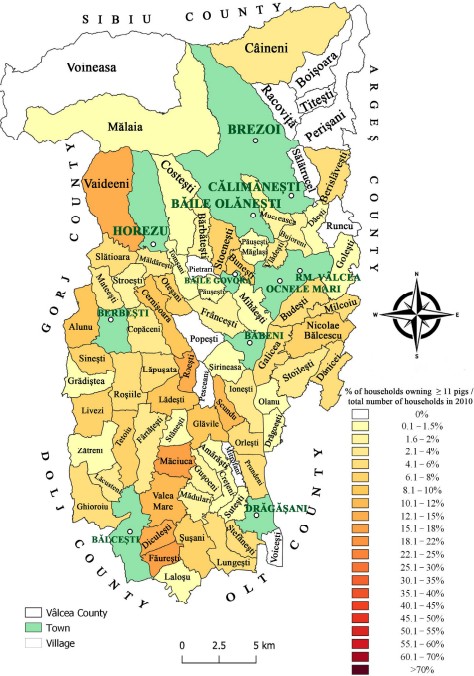

**Figure 25.** Share of households owning ≥11 pigs out of the total number of households in rural settlements in Vâlcea county in 2010. Source: elaborated by authors.

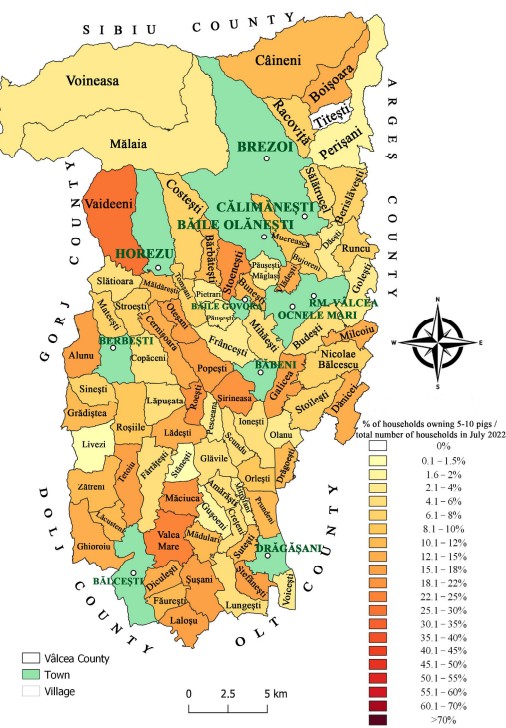

**Figure 26.** Share of households owning 5–10 pigs out of the total number of households in rural settlements in Vâlcea county in July 2022. Source: elaborated by authors.

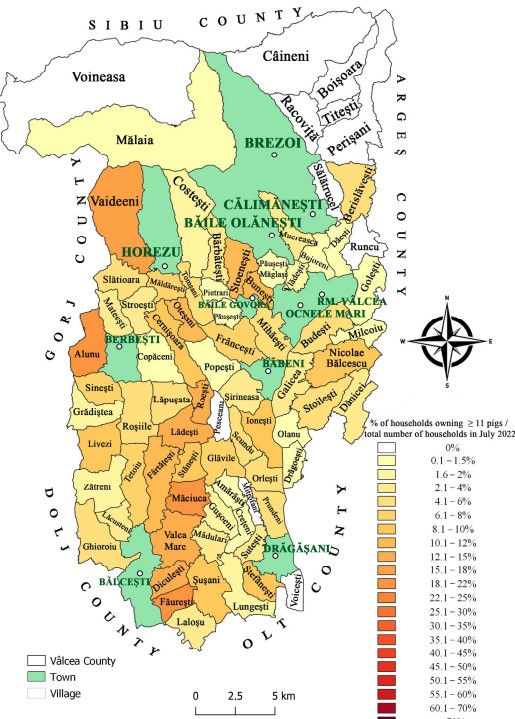

**Figure 27.** Share of households owning ≥11 pigs out of the total number of households in rural settlements in Vâlcea county in July 2022. Source: elaborated by authors.

The responses to questions about the reasons for raising pigs to satisfy meat consumption needs and increase revenue revealed the evolutive paradigm of this agricultural activity in the current period:

*"I have been raising pigs in my backyard for as long as I can remember. Now I have seven. Of them, four are for us, for the house and to give one to each of our children. That is the way it is with us. We offer pork to children because they made loans to buy their houses and they don't have enough money to pay for them. We, the parents, provide food. We sell the other three pigs. We get a certification from the vet, we verify meat and we get some money. Now we do not breed them as we used to, they have what they need. Cleanliness is compulsory, otherwise they get sick and we cannot do anything. If you want healthy animals with good meat, you have to offer them proper food and give them fresh, clean water several times a day"* (Viorica, 56 years old, Faureşti);

*"Here in our place raising pigs is the rule. Everybody, old or young, has at least one pig! I have nine. Of them, only two are for me; one for me and one for my son. He helps me when he can, because he works. The others (pigs) we sell. The money is good for me and for him. What is needed is to raise the animal properly! The food has to be natural and their space has to be clean. Their living space is not dirty and if I want to sell, I go to "the center" (the vet's office) to check the meat. I didn't cheat anyone!"* (Gheorghe, 66 years old, Lădeşti).

## 5. Discussion

This study focused on explaining the present distribution and concentration of pig farming according to existing demographic and economic factors. The authors analyzed the behavior of populations in rural areas in a county of Romania and swine breeding as a viable solution for households to satisfy their basic food needs and supplement their income. This study looked beyond Romanian cultural traditions, where parents help their children at any age through agricultural production, and similar results have been found in other studies [68]. For the current period, the aging of the population in the context of the financial crisis, unemployment, and job insecurity have changed the economic plans of some households, especially those with elderly people, whose incomes have been significantly affected by inflation. Each person and family is looking for ways to obtain the products needed to survive in this era of economic recession. These aspects were also emphasized by another study that focused on the national level [68].

Our research results show a clear dominance of small-scale farming and agriculture in CEE countries. In addition, they correlate with European statistics, which show that Romania has the largest number of small farms in the EU [69]. This situation may also indicate a growing interest in pursuing economic advantages [28], alongside the preservation of traditional occupations and social habits [70]. Raising animals on farms is a tradition. Besides the tradition of animal breeding for family consumption, there is a growing trend toward developing novel activities for growth based on traditional occupations such as pig farming.

In this context, stimulating economic activities is a priority in Romanian rural areas. A challenge that pig farming in Romania and the EU should consider in the future is the impact this activity would have on the environment given the need to increase meat production to meet demand while limiting the emissions generated by intensive animal breeding. These aspects were also emphasized in other studies [1,14,15,18,22,30–32,35,71]. European policies that recognize the animal agricultural sector as a significant source of non-CO2 greenhouse gases, along with the EU's Fit for 55 package that seeks to promote plant-based diets as a sustainable alternative to meat-based diets [72,73], would represent a major challenge to the future of pig farming and breeding in member countries such as Romania. The intensification of pork production and its promotion in external markets [9,39–42,54,55,59,74], as well as the introduction of innovative biogas plants, is feasible only for large farms [75] but could be a solution for sustainable planning in this agricultural sector.

The sustainability of the Romanian rural economy is closely linked to the potential to satisfy the consumption needs of the population through increasing and continuously diversifying products in line with the current and potential needs of consumers.

Pork is a primary product of local traditional gastronomy, with regionally specific recipes and denominations which are often impossible to translate into a foreign language (e.g., "rose" sausages, "carne la garnita" (smoked pork meat in lard), "sarmale" (tender sour cabbage leaves stuffed with ground pork, smoked bacon, and rice, slowly cooked for hours), and "peasant lard"). Many rural inhabitants, who are accustomed to raising pigs and are familiar with this occupation, satisfy their family's food consumption throughout the year while fulfilling a particular demand for traditional products considered a key part of Romanian culture. For the Christmas feast, some rural inhabitants will raise more than one or two pigs. This may be connected to the availability of food resources in certain regions (especially cereal-growing regions) that facilitate pig breeding. Additionally, there may be demands from family members and friends in urban areas for indigenous pork products, as well as opportunities to sell pork to supplement household revenues. This is consistent with other agricultural subsistence occupations with entrepreneurial initiatives in Romania [50].

As other studies have remarked, in the case of eastern Poland, home-prepared meat and "the practices connected with production, preparation and consumption of pork meat are key means of community building and functioning in the present reality" p. 223, Ref. [76]. This practice contributes to the construction of a social collective mentality, emphasizing a puzzling element of present territorial identity [77] and contributing to the preservation of local and regional traditional cultures, as well as ancient rural occupations in different regions in Romania. Therefore, the methodological framework and main results of this study could be of real interest to local authorities, civil servants with agricultural expertise, those representing institutions at different territorial levels, or other stakeholders (e.g., local and regional entrepreneurs, banks, etc.), particularly those interested in developing this agricultural domain and realizing its potential to ensure food safety and security in Romania in the future.

## 6. Conclusions

The results of this study can provide a perspective on future changes to the Romanian rural environment. These areas have a tradition of pig raising and other emergent areas for this occupation, which enables self-sufficiency in terms of meat consumption and supplements revenues for families involved in agriculture. Romanian agriculture faces a lot of uncertainties. Global climate change, economic and financial changes, energy crises, and consequences of the military conflict in Ukraine make the activities of small farmers difficult and force them to find alternative solutions such as pig raising and selling. Agricultural policies may help to improve the economic benefits of small farmers. Technology, marketing, consumer behavior, and development will always be advancing, and it is unlikely that any individual farmer alone can change these existing trends.

The results of this study validate the continuously evolving relationship between tradition and economic necessity for swine breeding in the rural environment of Vâlcea county. Here, the cultural character of swine breeding is less influenced by local economies and demographic characteristics. Population aging is not an obstacle to raising pigs in private households. The locals interviewed emphasized the role of this occupation in ensuring food self-sufficiency for the whole year. Our research also revealed that pig-breeding activities supplement revenues, particularly for socially vulnerable people such as retirees, unemployed, or unskilled workers. There were cases where the income obtained from the sale of pigs was the only source of revenue for the family.

This study also identified the microregional concentrations of swine breeding in Vâlcea county, emphasizing the areas with the potential to sell/buy pork from local farmers and areas in which the tradition of swine breeding for private consumption without commercialization activities prevails. Subsistence pig-raising activities in Romania also play a social role in reinforcing intergenerational relationships, driven by both cultural and economic factors. Parents raise pigs for themselves and their children and siblings,

maintaining strong, close social relationships between family members who live at a distance and often in different environments (e.g., rural farmers—urban consumers).

The utilized methodology helped to illustrate the territorial distribution, concentration, and evolution of this phenomenon, as it was the first time that this topic was approached from a geographical perspective in the studied territory. By combining quantitative statistics, illustrated through mapping techniques, with qualitative approaches through interviews, this topic provides valuable insights into the phenomenon of pig breeding and its complex implications. As such, it could be of significant interest to both scholars and practitioners seeking to understand this topic. This cultural and economic phenomenon is continuously evolving and increasing and is widespread in Romania. This study offers specific insights into this occupation for a representative region at a national level. The chosen topic, methodological framework, and area of study allow for the extrapolation of the overall results to larger regional and national scales. This enhances the value of the present research, making it of interest to a range of stakeholders (e.g., local administrations, agricultural planners, economic entrepreneurs) in a broader territorial context.

This study has certain limitations that may lead to further research on the topic. The most important limitation is the lack of statistical data at the local level on agricultural activities, which would have allowed more complex statistical analyses of this topic. In fact, most of the quantitative and qualitative data in this study were collected by the first author. Difficulties in obtaining relevant microscale data limited the quantitative analysis and the results of this research. Fortunately, these analyses were adequately completed using qualitative data, techniques, and results. The lack of continuous data on the essential basic indicators (e.g., pig counts at the local level—NUTS level 5) for the entire post-communist period (1990–2022) also limited the application of the cluster method, which may reveal more complex results if an extended statistical dataset becomes available in the future.

In this context, this research could provide a starting point for future and more in-depth studies. The approach of using different methodologies, including surveys on a stratified sample that take into account the types of farmers according to various criteria such as farm area and price adaptation, enhances our understanding of the complex phenomenon of pig breeding and its implications for local agricultural profitability. This approach allows us to describe patterns of behavior and consumption that are widespread in Romania, as well as in the nuanced European social context.

**Author Contributions:** Conceptualization, C.T.; data curation, C.T. and A.-I.L.-D.; formal analysis, C.T., M.B. and A.-I.L.-D.; investigation, C.T. and L.-Ş.S.; methodology, C.T. and M.B.; software, A.-N.J. and M.B.; validation, C.T., A.-I.L.-D and F.-C.M.; writing—original draft preparation C.T., A.-I.L.-D., M.B. and F.-C.M.; writing—review and editing, C.T., A.-I.L.-D. and F.-C.M.; supervision, C.T. and A.-I.L.-D.; project administration, C.T. All authors have read and agreed to the published version of the manuscript.

**Funding:** This research received no external funding.

**Institutional Review Board Statement:** This study was conducted in accordance with the guidelines of the Declaration of Helsinki and approved by the Ethics Committee of the Faculty of Geography, University of Bucharest.

**Informed Consent Statement:** Informed consent was obtained from all subjects involved in the study.

**Data Availability Statement:** The data presented in this study are available on request from the corresponding author. The data are not publicly available due to privacy and ethical restrictions.

**Conflicts of Interest:** The authors declare no conflict of interest.

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
