# Peer review of "Swine Breeding in the Villages of Vâlcea County, Oltenia (Romania)—Tradition or Necessity?"

_agriculture, doi:10.3390/agriculture13030733_

Round 1

Reviewer 1 Report (Previous Reviewer 1)

The article "Swine Breeding in the Villages of Vâlcea County, Oltenia (Romania) - Tradition or Necessity?" provides an up-to-date geographical analysis of the phenomenon of pig farming in Romania. The study employs a combination of methods, including cluster analysis and qualitative interviews, to reveal the region's complex motivations and spatial distribution of swine breeding. According to the study, pig breeding has cultural and economic significance, serving as a source of self-sufficiency and supplemental income for rural families. The research design is adequate, with direct interviews being especially effective. Overall, the article provides valuable information for scholars and practitioners interested in the changing cultural and economic implications of pig breeding in rural Romania.

Author Response

Dear Reviewer,

 Thank you very much for your appreciations and valuable comments that helped us improve our paper and obtain a better version of our manuscript.

All your comments and suggestions were considered and appropriate changes were brought and underlined with track changes in the manuscript.

Kind regards,

The Authors

Reviewer 2 Report (New Reviewer)

It is very important to keep a record of rural traditional customs.

This paper records the traditional practice of pig breeding in Romania.

I think it is a suitable study for this journal.

However, some proposals and amendments to the Plaintiff are required.

1. In the introduction, an Internet address was found. Change to a number, as in the journal (mdpi) regulations.
ex) (https://www.fao.org/3/cb5332en/Meat.pdf) -> [8]

2. Present a table on the number of pigs raised in the survey area (1, 2~4, 5~10, 11~). Include proportions in this table.

3. There are no statistical findings on Ukraine war. Suggest deleting the use of this word (reviewer's opinion).

4. There are no statistical findings for COVID-19. Suggest deleting the use of this word (reviewer's opinion).

5. Is it possible to correlate the number of families with the number of pigs raised?

6. How do Romanians feel about pigs?
-> Property? Family? Investment?
They eat pigs they used to raise like a family? - I can't say it's bad. - Find and refer to their feelings in movies, novels, etc.
-> This is similar to dog breeding in rural Korea in the past.

7. What is the main feed for pigs? Food waste?
- Link to number 5. - If you have a lot of family members, you can raise a lot of pigs due to a lot of food waste.

8. Let's focus on the traditional position and rural culture.

It will be a very important record. It's going to be hard, but please change it. If my suggestion is unfair, it can be rejected by your choice.

Author Response

Dear Reviewer,

Thank you very much for your appreciations and valuable comments that helped us improve our paper and obtain a better version of our manuscript.

All your comments and suggestions were considered and appropriate changes were brought and underlined with track changes in the manuscript.

Kind regards,

The Authors

This manuscript is a resubmission of an earlier submission. The following is a list of the peer review reports and author responses from that submission.

Round 1

Reviewer 1 Report

Dear Authors,

Congratulations on preparing an interesting article and using the very nice but seldom-used technique of mapping the result of the study. 

I find some areas to be improved:

1. In the abstract, you should also include the research question, a few words about methodology, and key findings.

2. Discussing pork consumption, it could be good to show some diagrams, for example, https://www.fao.org/faostat/en/#home. Moreover, there are some patterns in food consumption, like that there are similar food consumption structures, for example, in Poland, Romania, Hungary, and other post-communist countries (countries from the new EU) so forecasting the production and consumption trends you could use the data from that countries as background. 

3. The necessity vs. tradition would not be an issue in the coming decades from the perspective of decarbonization in the EU. Whether you are a fan of "Fit for 55" or not, You will have to consider that. To hack the policy, it could be useful to encourage establishing biogas facilities along with intensive swine production to lower the bad impact of the production on society.

Author Response

Dear Reviewer,

Thank you very much for your suggestions that we fully considered in order to improve our manuscript.

Please find our point-by-point response in the attached document.

Kind regards,

The Authors

Reviewer 2 Report

This paper focuses on pig farming in a representative NUTS2 administrative level of Romania by using mixed methods. Findings showed the concentration of swine breeding as a phenomenon, mapping its spreading for both subsistence and larger farms, while qualitative interviews better proved the motivation of farmers for this occupation. It also revealed areas of differing concentrations of both very small sized farms preoccupied with satisfying the self-necessities for food and larger farms which oriented their production in excess for commercialization and supplementing.

 I believe that this paper will bring an additional insight regarding the issue discussed in paper. Despite all the things, this needs some minor corrections are stated as follows:

         i.            Some comments should be added to the literature review section.

        ii.            It is suggested that the authors increase the basis for the application of method of cluster.

       iii.            The authors apply a different approach, how can different methods be integrated to reveal phenomena?

Otherwise, some minor mistakes:

         i.            Line 54, there are two commas.

        ii.            Line 412, the sentence is obscured by the picture.

Author Response

(The authors gave the same response as above.)

Reviewer 3 Report

“The activity of swine and animal breeding is aimed at meat consumption [1]. Meat 33 consumption, including pork, has reached notably high values [2], with the trends 34 reflecting a growing increase [3].”

The first paragraph of the introduction is very generic. Can you specify the context? Where is meat consumption increasing? What is the average per capita pork consumption (in Romania)?

“The present 37 research analyzes pig farming in Vâlcea county, a particular region of Romania which 38 displays this activity as part of the local culture, if considering its long tradition and 39 increasing trends for this occupation especially in rural settlements.”

Is pork from Valcea county characterised by a geographic indication (GI) or protected denomination of origin?

“At the same time, the growing number of pigs in rural farms is determined by 41 economic necessities [5]”

Can you provide data? What is the number of pigs? What is the number of farms?

When farmers choose to engage in pig farming, which reason takes precedent between the 92 tradition of swine breeding within the region and the economic motivation (earning income 93 from swine breeding within households in rural areas)?

The study appears to treat tradition and economic motivation as two distinct and mutually exclusive factors. (1) The contrast between the two factors is hardly explored or explained in the introduction. The factor of tradition in particular is barely mentioned. How is pig farming a tradition in the study location? (2) Can farmers or families not choose to engage in pig farming for both reasons?

“In order to further 240 extrapolate these possible behavior changes to the entire Romanian territory, the data 241 were analysed in four distinctive times in the post-socialist evolution: the 242 post-revolution moment (1990), EU pre-accession (2000), the Romanian economic crisis 243 (2010) and the contemporary period marked by the sanitary crisis and the Ukraine 244 invasion and war (2022).”

(1) Are you attempting to make a connection between these events and pig farming in Romania? If so, you need to elaborate on the supposed connections. (2) Is it not more likely that you conducted the study at predetermined intervals and these events just happened to occur at more or less the same time? (3) In 2010 most of the world experienced an economic/financial crisis, not just Romania. (4) What is the sanitary crisis? Swine flu?

“According to these norms, tradition is mainly the motivation for pig farming in the 266 rural settlements that register a large number of households opting for swine breeding in 267 the four times on which we focused our research (1990, 2000, 2010 and July 2022). The 268 presence of a large number of households with one pig or 2-4 pigs in certain settlements 269 show the existence of a local culture oriented on pig raising in private households for 270 self-consumption. Tradition is based on animal raising for private consumption. The 271 households with five or more pigs show a growing interest in raising pigs for commercial 272 purposes. In this case, economic necessities are expressed through commercialization 273 activities. In this way, the usefulness of this study is to differentiate areas in the studied 274 region (Vâlcea county) between those oriented towards swine farming for 275 self-consumption and those for commercialization.”

Based on this paragraph, the term “tradition” is just misused. Tradition has a historical connotation, which is not at all explored in the study. Instead, tradition is here confused with self-subsistence agriculture, which is the practice used by peasants and smallholders to produce food primarily for themselves. By using the number of pigs to categorise the farmers, the distinction is one between commercial and non-commercial farm operations.

Author Response

(The authors gave the same response as above.)
